

# Interferon-gamma and interleukin-10 genetic polymorphisms among the pulmonary tuberculosis caused by Malaysia-specific *Mycobacterium tuberculosis* strain: SIT745/EAI1-MYS

Nik Mohd Noor Nik Zuraina[1], Nur Annisa Nasuha Mohd Sedi[1], Mohammed Dauda Goni[2], Suharni Mohamad[3], Mohd Nor Norazmi[4,5] and Siti Suraiya[1,6]

[1] Department of Medical Microbiology & Parasitology, School of Medical Sciences, Universiti Sains Malaysia, Kubang Kerian, Kelantan, Malaysia
[2] Department of Veterinary Clinical Studies, Faculty of Veterinary Medicine, Universiti Malaysia Kelantan, Kota Bharu, Kelantan, Malaysia
[3] School of Dental Sciences, Universiti Sains Malaysia, Kubang Kerian, Kelantan, Malaysia
[4] School of Health Sciences, Universiti Sains Malaysia, Kubang Kerian, Kelantan, Malaysia
[5] Malaysian Genome and Vaccine Institute, National Institutes of Biotechnology Malaysia, Kajang, Selangor, Malaysia
[6] Hospital Infections and Epidemiology Control Unit, Hospital Pakar Universiti Sains Malaysia, Universiti Sains Malaysia, Kubang Kerian, Kelantan, Malaysia

Corresponding author
Siti Suraiya, ssuraiya@usm.my

## ABSTRACT

**Background:** *Mycobacterium tuberculosis* is the causative agent of tuberculosis, which results in 1.5 million global deaths annually. Key factors influencing tuberculosis development encompass host genetic factors, genetic diversity within *M. tuberculosis*, and environmental conditions. The East-African-Indian (EAI) lineage is the predominant lineage in Southeast Asia, India, and East Africa. A recent finding has identified SIT745/EAI1-MYS, a sub-lineage of EAI, as the geographically specific *M. tuberculosis* strain in Malaysia.

**Methods:** This study aimed to investigate the role of IFN-γ (+874) A/T and IL-10 (−1,082) A/G SNPs with tuberculosis (TB) disease susceptibility among patients infected with *M. tuberculosis* SIT745/EAI1-MYS strain. A cross-sectional study was conducted between three groups of subjects consisted of TB patients infected with *M. tuberculosis* SIT745/EAI1-MYS strain ($n = 9$), TB patients of non-*M. tuberculosis* SIT745/EAI1-MYS strains ($n = 9$), and healthy controls ($n = 9$). The genetic variation in IFN-γ (+874) A/T and IL-10 (−1,082) A/G SNPs were detected using allele-specific PCR and analyzed for their association with TB risk and severity.

**Results:** The results indicated a higher frequency of the IFN-γ (+874) TT and IL-10 (−1,082) AG genotypes among TB patients compared to healthy controls. The IFN-γ (+874) AA and IL-10 (−1,082) AG genotypes were more prevalent among TB patients infected with the SIT745/EAI1-MYS *M. tuberculosis* strains compared to non-SIT745/EAI1-MYS, indicating a possible link between these genotypes and more severe TB symptoms. Although there is no significant correlation between the IFN-γ (+874) A/T and IL-10 (−1,082) A/G polymorphisms with the susceptibility or severity of TB due to the small sample size, this initial strain-specific association

could suggest that genetic factors may interact with particular *M. tuberculosis* strains to influence disease severity.

# INTRODUCTION

Tuberculosis (TB) is a contagious airborne disease caused by *Mycobacterium tuberculosis*. It is among the deadliest diseases in the world, which caused approximately 1.3 million deaths in 2022, surpassing deaths due to human immunodeficiency virus/acquired immune deficiency syndrome (HIV/AIDS) in the same year (*World Health Organization, 2023*). The estimated annual rate of TB infection is nine million, out of which 1.5 million cases are fatal. Despite the high fatality rate associated with *M. tuberculosis*, only 5–10% of individuals who acquire the infection are prone to developing active TB. This small proportion suggests that susceptibility to TB is influenced by a combination of host genetic factors and environmental conditions (*Álvarez et al., 2023*). Additionally, research indicates that *M. tuberculosis* sub-lineages have evolved in diverse human populations, resulting in notable variations in virulence and immunomodulatory functions (*Saelens, Viswanathan & Tobin, 2019*). In Malaysia, the prevalent lineage is the East-African-Indian (EAI) lineage. A recent finding identified SIT745/EAI1-MYS, derived from the EAI lineage, as the geographically specific *M. tuberculosis* strain for Malaysia (*Ismail et al., 2014*).

The immune response to *M. tuberculosis* involves a delicate balance between pro-inflammatory and anti-inflammatory signals. Interleukin-10 (IL-10) and interferon-gamma (IFN-γ) are two important cytokines with diverse functions within the immune system and inflammation against TB. Additionally, they have the potential to serve as biomarkers for TB infection in diagnostic assessments for TB. IFN-γ is a key pro-inflammatory cytokine produced by natural killer (NK) cells and T-helper-1 (Th1) cells that mediates immunity to TB, especially by activating macrophages and other immune cells, including IL-10 and tumor necrosis factor-alpha (TNF-α), to kill *M. tuberculosis* (*De Martino et al., 2019*). It is also the key element in the containment of *M. tuberculosis* within the phagosome. The activation of macrophages leads to increased amounts of reactive oxygen and nitrogen species that causes oxidative damage to the bacterial cell. In addition, IFN-γ inhibits Th2 (T-helper 2) immune responses that are associated with anti-inflammatory and tissue repair phenotype to maintain a balanced immune response and prevents excessive tissue damage during infection (*De Martino et al., 2019*).

Meanwhile, IL-10 is an anti-inflammatory cytokine that inhibits the synthesis of several pro-inflammatory cytokines, including IFN-γ (*Adane et al., 2021*). IL-10 is also involved in the formation and regulation of granuloma that is composed by immune cells such as lymphocytes, macrophages, epithelioid cells and fibroblasts, in response to *M. tuberculosis*

infections. As an inhibitory cytokine, IL-10 plays a crucial role in the downregulation of Th1 cytokines, MHC class II antigen expression, and co-stimulatory molecular expression on macrophages, which helps in maintaining a balanced immune environment. While this immunosuppressive function may help limit excessive inflammation and tissue damage, it can also hinder the effective elimination of the bacteria and impair the ability of macrophages to effectively kill and clear the bacteria (*Wani et al., 2021*).

Different genetic polymorphisms which modulate the host immune response in favour of TB infection and disease progression have been identified in human cytokines. Single nucleotide polymorphisms (SNPs) in genes encoding IFN-γ and IL-10 can influence the susceptibility to and the progression of TB (*Adane et al., 2021*). The human IFN-γ is located on chromosome 12 (12q14) and has four exons spanning about 6 kb. The initial intron of this IFN-γ gene contains the binding site for the transcription factor NFκB, a key regulator that is involved in the expression of IFN-γ and other cytokine genes. An SNP at the position +874 A>T (rs2430561) of this intron is hypothesized to affect the gene expression and cytokine secretion, thereby exerting a notable influence on the outcome of infections (*Areeshi et al., 2021*). Studies have indicated that TB patients who have the homozygous allele A combination exhibit markedly lower levels of IFN-γ production compared to those carrying one or two copies of the allele T (*Areeshi et al., 2021*). Similarly, the SNPs in IL-10 −1,082 A>G may lead to increased production of IL-10 that will suppress the immune response and consequently allow the persistence of *M. tuberculosis*. Moreover, SNPs may result in the imbalance between pro-inflammatory and anti-inflammatory responses, which may compromise the host's ability to mount an effective immune response against the bacteria. Genetic variations in IL-10 may also influence the formation and maintenance of granulomas, affecting the containment of the bacteria.

Malfunction in the IFN-γ and IL-10 genes may undermine the host's capability to mount an efficient response to *M. tuberculosis* infection, heightening susceptibility to the disease and increasing the likelihood of developing active tuberculosis. Recent case-control studies have analyzed the association between SNPs in IFN-γ +874 A>T and IL-10 −1,082 G>A genes and TB development in diverse populations. However, the results of these studies have been inconsistent, leaving uncertainty about whether this polymorphism is linked to an increased or decreased susceptibility to TB. Therefore, this study aimed to investigate the association of IFN-γ +874 A/T and IL-10 −1,082 G>A SNPs with TB disease susceptibility among patients infected with the Malaysia-specific *M. tuberculosis* strain, SIT745/EAI1-MYS.

## MATERIALS AND METHODS

### Subjects

This study involved three groups of subjects: TB patients infected with *M. tuberculosis* strain SIT745/EAI1-MYS, TB patients infected with *M. tuberculosis* from other lineages (non-SIT745/EAI1-MYS), and healthy controls. Patients in the SIT745/EAI1-MYS group were specifically selected through initial screening of *M. tuberculosis* isolates. Archived *M. tuberculosis* isolates from the years 2011 to 2022 were screened using PCR-based typing

to identify unique spacers defining the isolates as SIT745/EAI1-MYS (*Ismail et al., 2014*). Patients belonging to SIT745/EAI1-MYS isolates were traced and identified using the Laboratory Information System (LIS) of Hospital Universiti Sains Malaysia (USM). Similar procedures were carried out for subjects in the non-SIT745/EAI1-MYS group. Meanwhile, the healthy control group consisted of individuals with no history of tuberculosis. All subjects in this study were of Kelantanese Malay ethnicity and had no history of HIV positivity. Following the acquisition of written consent, demographic and clinical data were collected and recorded. A buccal swab was collected from each subject by rubbing a sterile cotton swab inside the cheeks for 20 s. Three buccal swab samples were collected from each subject and immediately suspended in a microcentrifuge tube containing 1 ml of 0.9% normal saline, then stored at 4 °C until used for DNA extraction. Research was conducted in the Research Laboratory, Department of Medical Microbiology and Parasitology, USM. This study was approved by the Human Research Ethics Committee, USM (Reference number: USM/JEPeM/18080362).

## DNA extraction from buccal swab

DNA extraction from buccal swab samples was carried out using the QiaAmp DNA Mini kit (Qiagen, Hilden, Germany). Initially, the normal saline solution containing buccal mucosa cells was mixed by vortexing for 15 s and 400 µL of the cell suspension was added to AL lysis buffer in 1:1 ratio. The mixture was added with 20 µL of proteinase K solution and incubated at 56 °C for 15 min. After mixing with 400 µL of absolute ethanol, the whole mixture was transferred gradually to a spin column, underwent centrifugation and proceed with the remainder washing steps in accordance to the manufacturer's guidelines. DNA was eluted using 50 µL of AE buffer and stored at −20 °C for molecular genotyping analysis.

## Molecular genotyping by allelic-specific PCR

Molecular genotyping using allelic-specific PCR was performed to study the characteristics of genetic polymorphisms in IFN-γ (+874 A/T) and IL-10 (−1,082 A/G) genes separately.

To identify the polymorphism at +874 A/T of IFN-γ, two distinct reactions were established for each target. One tube contained a primer specific for the 'A' allele, while the other tube contained a primer specific for the 'T' allele. The presence or absence of bands at 261 bp for IFN-γ in these two tubes was noted. A parallel procedure was implemented for IL-10 (−1,082 A/G) allelic PCR, targeting a size of 550 bp. An internal control of size 426 bp was incorporated to verify the successful functionality of the PCR. The primers used were listed in Table 1.

A total volume of 25 µL PCR reaction mixture contained 5 µL of MyTaq Red reaction buffer (1×) and 0.3 µL of MyTaq Red DNA polymerase (0.5 units) (Bioline, Taunton, MA, USA), 1 µL of each sense and antisense primer (1 µM), 4.7 µL of dH$_2$O, and 3 µL of DNA template. The PCR reaction consisted of initial denaturation at 95 °C for 1 min, 10 cycles of denaturation at 95 °C for 15 s, annealing at 62 °C for 50 s, and elongation at 72 °C for 40 s, followed by 20 cycles 95 °C for 20 s, 56 °C for 50 s, and 72 °C for 50 s. An internal control

**Table 1 List of primers used in the molecular genotyping of IFN-γ (+874 A/T) and IL-10 (−1,082 A/G) genes.**

| Target amplicon | Sequence (5′-3′) | Amplicon size (bp) | Remark |
|---|---|---|---|
| IFN-γ 874*A | F: TTCTTACAACACAAAATCAAATCA | 261 | IFN-γ (+874) A/T |
| | R: TCAACAAAGCTGATACTCCA | | |
| IFN-γ 874*T | F: TTCTTACAACACAAAATCAAATCT | 261 | |
| | R: TCAACAAAGCTGATACTCCA | | |
| IL-10 −1,082*A | F: CTACTAAGGCTTCTTTGGGAA | 550 | IL-10 (−1,082) A/G |
| | R: CAGCCCTTCCATTTTACTTTC | | |
| IL-10 −1,082*G | F: CTACTAAGGCTTCTTTGGGAG | 550 | |
| | R: CAGCCCTTCCATTTTACTTTC | | |
| Human growth hormone gene | F: GCCTTCCAACCATTCCCTTA | 426 | Internal control |
| | R: TCACGGATTTCTGTTGTGTTTC | | |

of 426 bp was amplified using a pair of primers designed from the nucleotide sequence of the human growth hormone.

## Statistical analysis

Data processing and analysis were conducted *via* statistical software SPSS version 24.0. The genotype polymorphisms between the control and TB patient groups were compared using the chi-squared test or Fisher's exact test, as appropriate. When more than 20% of the expected counts are less than 5, the Fisher's exact test was used instead of the Chi-squared test. Relative risk and odds ratio were calculated to assess disease susceptibility and clinical course. A $p$ value $< 0.05$ was considered statistically significant. Due to the small sample size, *post hoc* effect size calculations for Cohen's $w$ and power analyses were also conducted based on the chi-squared test results to evaluate the strength of association and statistical robustness.

## RESULTS

### Demographic and clinical characteristics of the study participants

A total of 18 subjects, consisting of nine individuals from each group, consented to participate in this study. Table 2 shows the demographic data of TB patients and healthy control groups. Within the TB-SIT745/EAI1-MYS group, the majority were males (78%), aged less than 50 years (67%), and active smokers (44%). In contrast, the TB-non-SIT745/EAI1-MYS group primarily consisted of females (77.8%), aged over 50 years (67%), and non-smokers (78%). In this study, TB patients were included regardless of their diabetes mellitus (DM) status, due to the notably high prevalence among the Malay population in Kelantan, where the research was conducted. The inclusion of type 2 DM-positive individuals was deemed appropriate to ensure that the study reflects the real-world patient population in this region. Although DM may serve as a potential confounding factor in immunogenetic studies, the comparable proportion of DM-positive subjects in both TB-Malaysian strain and TB-non-Malaysian strain groups (67%) could reduce the risk of confounding bias for a more reliable analyses of genetic polymorphisms between these

**Table 2 Demographic and clinical characteristics of the subjects.**

| Characteristics | Study groups | | | p value |
| --- | --- | --- | --- | --- |
| | Healthy control n = 9 (%) | TB-SIT745/EAI1-MYS n = 9 (%) | TB-Non-SIT745/EAI1-MYS, n = 9 (%) | |
| Gender | | | | |
| Male | 4 (44.4) | 7 (77.8) | 2 (22.2) | 0.053 |
| Female | 5 (55.6) | 2 (22.2) | 7 (77.8) | |
| Age groups | | | | |
| <50 | 6 (66.7) | 3 (33.3) | 4 (44.4) | 0.403 |
| >50 | 3 (33.3) | 6 (66.7) | 5 (55.6) | |
| Smoking history | | | | |
| Active smoker | 1 (11.1) | 4 (44.4) | 1 (11.1) | 0.018 |
| Ex-smoker | 0 (0.0) | 3 (33.3) | 1 (11.1) | |
| Non-smoker | 8 (88.9) | 2 (22.2) | 7 (77.8) | |
| Comorbidities | | | | |
| NKMI | 9 (100.0) | 3 (33.3) | 3 (33.3) | 0.010 |
| DM | 0 (0.0) | 6 (66.7) | 6 (66.7) | |
| HPT | 0 (0.0) | 4 (44.4) | 6 (66.7) | |
| KD | 0 (0.0) | 1 (11.1) | 2 (22.2) | |
| Year of TB infection | | | | |
| 2011–2016 | 0 (0.0) | 2 (22.2) | 1 (11.1) | 0.667 |
| 2017–2021 | 0 (0.0) | 7 (77.8) | 8 (88.9) | |

Note:
NKMI, no known medical illness; DM, diabetes mellitus; HPT, hypertension; KD, kidney diseases; $p < 0.05$ indicates statistical significance.

groups. On the other hand, hypertension was notably higher in the TB-non-SIT745/EAI1-MYS group (67%). The control group comprised healthy individuals with no tuberculosis history, with a slightly higher proportion of females (56%), individuals aged less than 50 years (56%), and the majority being non-smokers (89%).

## Genetic polymorphisms in IFN-γ (+874 A/T) and IL-10 (−1,082 A/G) genes

Genetic polymorphisms in the IFN-γ (+874 A/T) and IL-10 (−1,082 A/G) genes were observed in all three groups, based on the presence or absence of the target amplicons (Fig. 1). For IFN-γ (+874 A/T), the presence of bands at 261 bp in both tubes indicated the AT genotype, while the presence of bands in only the first or second tube at 261 bp represented the AA genotype or the 'TT' genotype, respectively. The same pattern of observation was applied to the 550 bp target bands of IL-10 (−1,082 A/G). The results for IFN-γ (+874 A/T) and IL-10 (−1,082 A/G) genotype profiles were summarized in Table 3.

The correlation between IFN-γ +874 and IL-10 −1,082 genetic polymorphism and the odds of developing TB were analyzed as shown in Table 4. More than half (55.6%) of the TB patients infected with *M. tuberculosis* strains SIT745/EAI1-MYS were observed to carry +874 AA genotypes of IFN-γ, similar proportion to that of the healthy control group. The

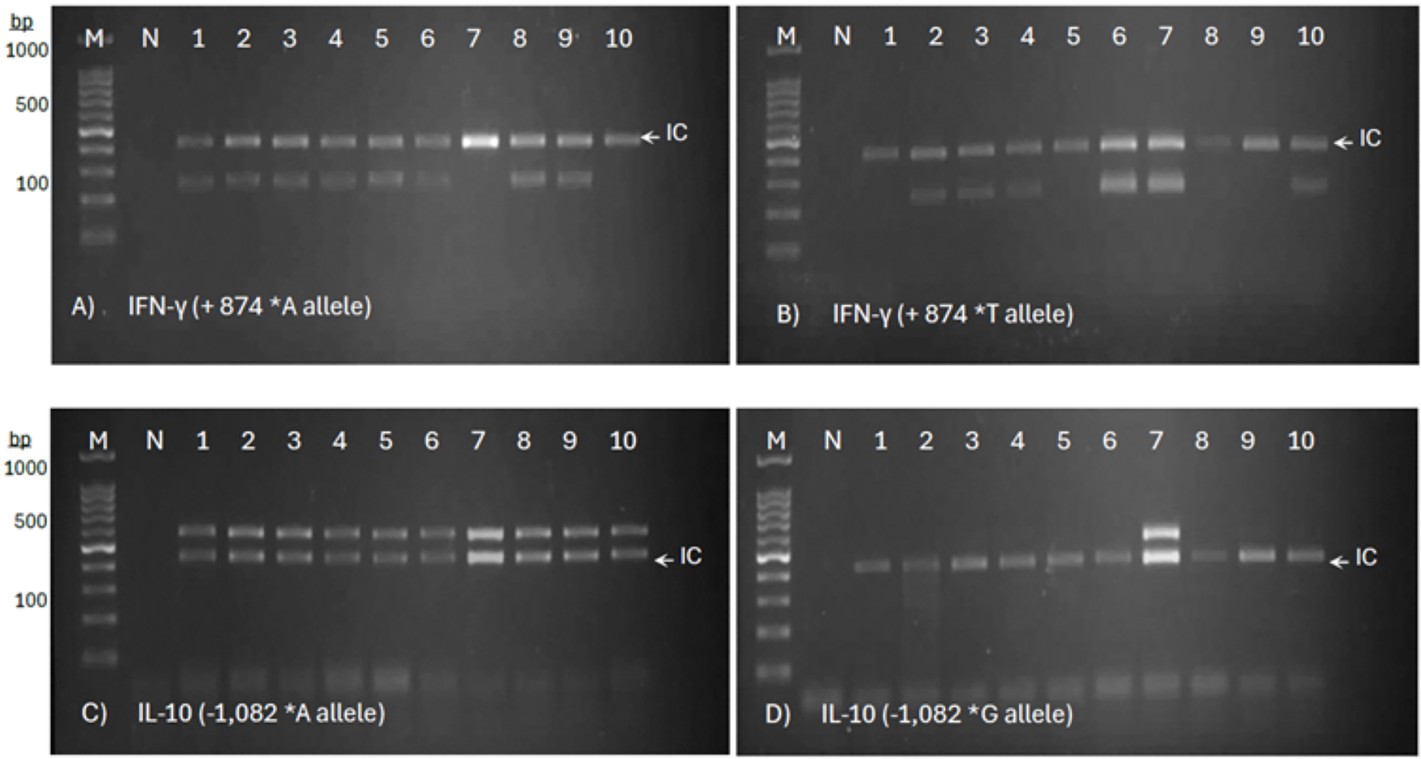

**Figure 1 Allele-specific PCR genotyping of IFN-γ (+874 A/T) and IL-10 (−1,082 A/G) SNPs in PTB patients.** The representative agarose gel images show a two-tube reaction system for allele-specific PCR, with primer sets targeting the 'A' allele (A) and 'T' allele (B) of IFN-γ, and the 'A' allele (C) and 'G' allele (D) of IL-10. The presence of a distinct amplicon band indicates successful amplification of the respective allele.

frequency of IFN-γ +874 A allele was higher than T allele in all three groups, including both TB patients and healthy controls. Among all, healthy controls carried a higher proportion of A allele, while patients infected with non-SIT745/EAI1-MYS strains carried the highest proportion of T allele. Interestingly, none of the healthy controls carried IFN-γ +874 TT homozygote genotype. This may indicate that individuals with IFN-γ +874 T allele might be more susceptible to tuberculosis infection. The distribution of genotypic and allelic frequencies of IFN-γ +874 A/T SNP in TB patients and healthy controls was not statistically significant (OR, 1.56; 95% CI [0.31–7.82]; $p$ = 0.587) and (OR, 2.27; 95% CI [0.61–8.15]; $p$ = 0.227).

Meanwhile, it was observed that the occurrence of the IL-10 (−1,082) AA genotype was 100% among healthy controls, 66.7% among SIT745/EAI1-MYS TB patients, and 77.8% among non-SIT745/EAI1-MYS TB patients. The heterozygous AG genotype was observed in 33.3% and 22.2% of the patients infected with SIT745/EAI1-MYS and non- SIT745/EAI1-MYS, respectively. None of the participants from all the three groups of this study carry the GG genotype. There are no significant differences in genotypic and allelic frequencies of IL-10 (−1,082 A/G) in all three groups (OR, 7.74; 95% CI [0.38–157.31]; $p$ = 0.183) and (OR, 6.46; 95% CI [0.34–123.61]; $p$ = 0.215).

**Table 3 The summary of allele-specific PCR genotyping of IFN-γ (+874 A/T) and IL-10 (−1,082 A/G) SNPs in all three subject groups.**

| No. | Subjects | Group description | Age | Gender | Allelic PCR Target: IFN-γ +874 A/T (261 bp) | | | | | Allelic PCR Target: IL-10 -1,082 A/G (550 bp) | | | | |
|---|---|---|---|---|---|---|---|---|---|---|---|---|---|---|
| | | | | | IC (426 bp) | IFN-γ +874*A | IC (426 bp) | IFN-γ +874*T | Genotypes | IC (426 bp) | IL-10 −1,082*A | IC (426 bp) | IL-10 −1,082*G | Genotypes |
| 1 | G1 (C1) | HC | 24 | F | ✓ | ✓ | ✓ | - | AA | ✓ | ✓ | ✓ | - | AA |
| 2 | G1 (C2) | HC | 33 | F | ✓ | ✓ | ✓ | ✓ | AT | ✓ | ✓ | ✓ | - | AA |
| 3 | G1 (C3) | HC | 30 | F | ✓ | ✓ | ✓ | - | AA | ✓ | ✓ | ✓ | - | AA |
| 4 | G1 (C4) | HC | 29 | F | ✓ | ✓ | ✓ | ✓ | AT | ✓ | ✓ | ✓ | - | AA |
| 5 | G1 (C5) | HC | 41 | F | ✓ | ✓ | ✓ | - | AA | ✓ | ✓ | ✓ | - | AA |
| 6 | G1 (C6) | HC | 45 | M | ✓ | ✓ | ✓ | ✓ | AT | ✓ | ✓ | ✓ | - | AA |
| 7 | G1 (C7) | HC | 53 | M | ✓ | ✓ | ✓ | - | AA | ✓ | ✓ | ✓ | - | AA |
| 8 | G1 (C8) | HC | 60 | M | ✓ | ✓ | ✓ | ✓ | AT | ✓ | ✓ | ✓ | - | AA |
| 9 | G1 (C10) | HC | 60 | M | ✓ | ✓ | ✓ | - | AA | ✓ | ✓ | ✓ | - | AA |
| 10 | G2 (P1) | MY TB | 62 | M | ✓ | ✓ | ✓ | ✓ | AT | ✓ | ✓ | ✓ | - | AA |
| 11 | G2 (P2) | MY TB | 46 | M | ✓ | ✓ | ✓ | - | AA | ✓ | ✓ | ✓ | - | AA |
| 12 | G2 (P3) | MY TB | 66 | M | ✓ | - | ✓ | ✓ | TT | ✓ | ✓ | ✓ | - | AA |
| 13 | G2 (P4) | MY TB | 68 | M | ✓ | ✓ | ✓ | - | AA | ✓ | ✓ | ✓ | - | AA |
| 14 | G2 (P5) | MY TB | 71 | M | ✓ | - | ✓ | ✓ | TT | ✓ | ✓ | ✓ | - | AA |
| 15 | G2 (P6) | MY TB | 39 | F | ✓ | ✓ | ✓ | - | AA | ✓ | ✓ | ✓ | ✓ | AG |
| 16 | G2 (P7) | MY TB | 31 | F | ✓ | ✓ | ✓ | ✓ | AT | ✓ | ✓ | ✓ | ✓ | AG |
| 17 | G2 (P9) | MY TB | 61 | M | ✓ | ✓ | ✓ | – | AA | ✓ | ✓ | ✓ | ✓ | AG |
| 18 | G2 (P10) | MY TB | 60 | M | ✓ | ✓ | ✓ | – | AA | ✓ | ✓ | ✓ | – | AA |
| 19 | G3 (X1) | Non-MY TB | 24 | M | ✓ | ✓ | ✓ | – | AA | ✓ | ✓ | ✓ | – | AA |
| 20 | G3 (X2) | Non-MY TB | 66 | F | ✓ | ✓ | ✓ | ✓ | AT | ✓ | ✓ | ✓ | - | AA |
| 21 | G3 (X3) | Non-MY TB | 68 | F | ✓ | ✓ | ✓ | ✓ | AT | ✓ | ✓ | ✓ | - | AA |
| 22 | G3 (X4) | Non-MY TB | 55 | F | ✓ | ✓ | ✓ | ✓ | AT | ✓ | ✓ | ✓ | - | AA |
| 23 | G3 (X5) | Non-MY TB | 25 | F | ✓ | ✓ | ✓ | - | AA | ✓ | ✓ | ✓ | ✓ | AG |
| 24 | G3 (X6) | Non-MY TB | 26 | F | ✓ | ✓ | ✓ | ✓ | AT | ✓ | ✓ | ✓ | - | AA |
| 25 | G3 (X7) | Non-MY TB | 56 | M | ✓ | - | ✓ | ✓ | TT | ✓ | ✓ | ✓ | ✓ | AG |
| 26 | G3 (X9) | Non-MY TB | 49 | F | ✓ | ✓ | ✓ | - | AA | ✓ | ✓ | ✓ | - | AA |
| 27 | G3 (X10) | Non-MY TB | 55 | F | ✓ | - | ✓ | ✓ | TT | ✓ | ✓ | ✓ | - | AA |

**Note:**
IC, Internal control PCR amplification; HC, healthy control subjects; MY TB, patients infected with Malaysian *M. tuberculosis* strain; Non-MY TB, patients infected with other *M. tuberculosis* strains; F, female; M, male; ✓, presence of target amplicon; -, absence of target amplicon band.

To assess the statistical robustness of the observed associations despite the small sample size, *post hoc* effect size calculations and power analyses were performed. The calculated Cohen's *w* for IFN-γ (+874 A/T) and IL-10 (−1,082 A/G) polymorphisms were 0.53 and 0.81, respectively, which indicate large effect sizes based on Cohen's criteria (*Ben Shachar et al., 2023*). The large effect sizes suggest that the observed genetic variations may have meaningful associations with TB susceptibility in a broader population. Additionally, *post hoc* power analysis $(1 - \beta)$ revealed moderate to high statistical power for the observed differences, with the power values of 0.65 for IFN-γ (+874 A/T) and 0.88 for IL-10 (−1,082 A/G). These findings support the relevance of the observed genotypes distributions despite

**Table 4 The association between IFN-γ +874 and IL-10 −1,082 genetic polymorphism and the odds of developing TB.**

| Genotypes | Healthy control | TB-SIT745/EAI1-MYS | TB-Non-SIT745/EAI1-MYS | Chi-square (p value) | Healthy control vs TB patients OR (95% CI)/p value |
|---|---|---|---|---|---|
| IFN-γ (+874 A/T) | n = 9 (%) | n = 9 (%) | n = 9 (%) | 2.44 (0.418) | 1.563[a] |
| AA | 5 (55.6) | 5 (55.6) | 3 (33.3) | | (0.312 to 7.819)/ |
| AT | 4 (44.4) | 2 (22.2) | 4 (44.4) | | 0.587 |
| TT | 0 (0.0) | 2 (22.2) | 2 (22.2) | | |
| Allele A/T | n = 18 (%) | n = 18 (%) | n = 18 (%) | | 2.227[b] |
| A | 14 (77.8) | 12 (66.7) | 10 (55.6) | | (0.608 to 8.153)/ |
| T | 4 (22.2) | 6 (33.3) | 8 (44.4) | | 0.227 |
| IL-10 (−1,082 A/G) | n = 9 (%) | n = 9 (%) | n = 9 (%) | 3.38 (0.128) | 7.741[c] |
| AA | 9 (100.0) | 6 (66.7) | 7 (77.8) | | (0.381 to 157.314)/ |
| AG | 0 (0.0) | 3 (33.3) | 2 (22.2) | | 0.183 |
| GG | 0 (0.0) | 0 (0.0) | 0 (0.0) | | |
| Allele A/G | n = 18 (%) | n = 18 (%) | n = 18 (%) | | 6.460[d] |
| A | 18 (100.0) | 15 (83.3) | 16 (88.9) | | (0.338 to 123.614)/ |
| G | 0 (0.0) | 3 (16.7) | 2 (11.1) | | 0.215 |

Notes:
[a] Odds ratio for IFN-γ (+874) AT/TT vs AA genotypes.
[b] Odds ratio for IFN-γ (+874) T vs A alleles.
[c] Odds ratio for IL-10 (−1,082) AG vs AA.
[d] Odds ratio for IL-10 (−1,082) G vs A alleles
$p < 0.05$ indicates statistical significance.

the limited sample size, and underscore the importance of further studies with larger cohorts to validate these preliminary associations.

## Analysis of TB severity between patients infected with SIT745/EAI1-MYS and non-SIT745/EAI1-MYS *M. tuberculosis* strains

The severity of TB was initially compared between patients infected with SIT745/EAI1-MYS and non-SIT745/EAI1-MYS *M. tuberculosis* strains based on the signs and symptoms as shown in Table 5. Overall findings demonstrated more or less similar degree of severity in both groups. Prolonged cough of more than two months was common for all the patients in both groups. However, a higher proportion of patients infected with SIT745/EAI1-MYS were observed to experience night sweats (77.8%) and loss of weight (77.8%) in comparison to non- SIT745/EAI1-MYS strains. Based on the chest radiograph reports, two patients from *M. tuberculosis* SIT745/EAI1-MYS infection had bilateral consolidation, whereas, other TB patients from both groups were observed to have unilateral consolidation. For TB patients infected with non-SIT745/EAI1-MYS strains, the symptoms for chest pain, hemoptysis, and shortness of breath were higher than those in the other group.

The scale for TB severity was assigned into mild, moderate, and severe, based on the types and the number of symptoms presence in each patient. Patients presented with less than four symptoms, excluding the hemoptysis and bilateral chest consolidation, were

**Table 5 Analysis of TB severity between patients infected with SIT745/EAI1-MYS and non-SIT745/EAI1-MYS *M. tuberculosis* strains.**

| TB symptoms | TB patients | | p value |
|---|---|---|---|
| | SIT745/EAI1-MYS strains, $n = 9$ (%) | Non-SIT745/EAI1-MYS strains, $n = 9$ (%) | |
| **On-off Fever** | | | |
| Yes | 6 (66.7) | 9 (100.0) | 0.206 |
| No | 3 (33.3) | 0 (0.0) | |
| **Persistent cough** | | | |
| Yes | 9 (100.0) | 9 (100.0) | – |
| No | 0 (0.0) | 0 (0.0) | |
| **Night sweats** | | | |
| Yes | 7 (77.8) | 0 (0.0) | 0.002 |
| No | 2 (22.2) | 9 (100.0) | |
| **Loss of weight** | | | |
| Yes | 7 (77.8) | 5 (55.6) | 0.620 |
| No | 2 (22.2) | 4 (44.4) | |
| **Hemoptysis** | | | |
| Yes | 1 (11.1) | 2 (22.2) | 0.500 |
| No | 8 (88.9) | 7 (77.8) | |
| **Shortness of breath** | | | |
| Yes | 3 (33.3) | 5 (55.6) | 0.637 |
| No | 6 (66.7) | 4 (44.4) | |
| **Chest pain** | | | |
| Yes | 0 (0.0) | 2 (22.2) | 0.471 |
| No | 9 (100.0) | 7 (77.8) | |
| **Sputum AFB smear index** | | | |
| +++ | 3 (33.3) | 3 (33.3) | 1.000 |
| ++ | 1 (11.1) | 1 (11.1) | |
| + | 5 (55.6) | 5 (55.6) | |
| **Chess radiograph** | | | |
| **Laterality** | | | |
| Unilateral | 7 (77.8) | 9 (100.0) | 0.224 |
| Bilateral | 2 (22.2) | 0 (0.0) | |
| **TB severity scale** | | | |
| Mild | 3 (33.3) | 3 (33.3) | 1.000 |
| Moderate | 3 (33.3) | 4 (44.4) | |
| Severe | 3 (33.3) | 2 (22.2) | |

Note:
AFB, Acid fast bacilli; +++, 10 AFB per field (check 20 fields); ++, 1–10 AFB per field (check 50 fields); +, 10–99 AFB in 100 fields; $p < 0.05$ indicates statistical significance.

grouped into mild TB. Those with more than three symptoms and in the presence of shortness of breath were classified as moderate. Severe TB group comprised any patients presented with hemoptysis and bilateral parenchymal consolidation. Patients infected with

SIT745/EAI1-MYS strain were recorded to have higher rate for severe TB (33.3%) compared to the non- SIT745/EAI1-MYS strains (22.2%).

## Association of IFN-γ (+874 A/T) SNPs with demographic and clinical parameters in healthy controls and tuberculosis patients infected with SIT745/EAI1-MYS and non-SIT745/EAI1-MYS *M. tuberculosis* strains

The association of IFN-γ (+874 A/T) SNPs with demographic and clinical parameters were analyzed (Table 6). Overall, none of the healthy control and young participants of aged less than 50 years from both TB groups carry TT genotype. More than 77% of the TB patients were recorded with recent TB infections between the year 2017 and 2021. More than 60% of the patients who presented with shortness of breath were found to carry AT genotype.

More than half of the TB patients in the SIT745/EAI1-MYS group were IFN-γ (+874) AA genotype carriers ($n = 5/9$, 56%). Among the male patients in this groups, 57% ($n = 4/7$) carry the AA genotype, all of whom had underlying diabetes mellitus (DM). All the five AA genotype carriers in this group experienced weight loss, a majority of them ($n = 4/5$, 80%) had a low AFB smear index, which indicated a low amount of tuberculosis bacilli in their sputum specimens, and 80% ($n = 4/5$) presented with unilateral consolidation chest radiographs. None of them were recorded with shortness of breath. However, two patients with the AA genotype were classified as having severe TB due to hemoptysis and bilateral consolidation, respectively. Apart from the AA carriers, the SIT745/EAI1-MYS group included two individuals with IFN-γ (+874) AT genotype (22%) and two with TT genotype (22%). Both AT carriers had no underlying comorbidities and were non-active smokers, but they presented with shortness of breath, fever, loss of weight, and night sweats. One of these AT carriers had severe TB infections with bilateral consolidation in the chest radiograph ($n = 1/2$, 50%). The two patients with TT genotype in this group were both males aged over 50, active smokers, and had underlying DM. Both of the patients did not exhibit severe TB signs and symptoms.

On the contrary, almost half of the patients in the non-SIT745/EAI1-MYS group were IFN-γ (+874) AT carriers ($n = 4/9$, 45%), and all of them were female, non-smokers, and had recently acquired TB infections. These patients did not exhibit night sweats or bilateral consolidation laterality, and none of them had a third-degree AFB smear index. However, one of the patients was documented with hemoptysis and classified as having a severe TB infection ($n = 1/4$, 25%). Meanwhile, all three AA genotype carriers in this group were young TB patients (under 50 years old), and none of them had hemoptysis or bilateral consolidation chest findings. As for the two patients with the TT genotype in this group, both were over 50 years old and had underlying DM and HPT. Notably, one of these TT genotype carriers experienced hemoptysis and was recorded as having severe TB ($n = 1/2$, 50%).

**Table 6 Association of IFN-γ (+874 A/T) SNPs with demographic and clinical parameters in healthy controls and tuberculosis patients infected with SIT745/EAI1-MYS and non-SIT745/EAI1-MYS *M. tuberculosis* strains.**

| Characteristics | Healthy control | | | | | TB-SIT745/EAI1-MYS | | | | | TB-Non-SIT745/EAI1-MYS | | | | |
|---|---|---|---|---|---|---|---|---|---|---|---|---|---|---|---|
| | $n = 9$ | AA | AT | TT | $p$ value | $n = 9$ | AA | AT | TT | $p$ value | $n = 9$ | AA | AT | TT | $p$ value |
| **Gender** | | | | | | | | | | | | | | | |
| Male | 4 | 2 | 2 | 0 | 0.956 | 7 | 4 | 1 | 2 | 0.477 | 2 | 1 | 0 | 1 | 0.325 |
| Female | 5 | 3 | 2 | 0 | | 2 | 1 | 1 | 0 | | 7 | 2 | 4 | 1 | |
| **Age groups** | | | | | | | | | | | | | | | |
| <50 | 6 | 3 | 3 | 0 | 0.894 | 3 | 2 | 1 | 0 | 0.509 | 4 | 3 | 1 | 0 | 0.051 |
| >50 | 3 | 2 | 1 | 0 | | 6 | 3 | 1 | 2 | | 5 | 0 | 3 | 2 | |
| **Smoking history** | | | | | | | | | | | | | | | |
| Active smoker | 1 | 0 | 1 | 0 | 0.843 | 4 | 2 | 0 | 2 | 0.406 | 1 | 1 | 0 | 0 | 0.199 |
| Ex-smoker | 0 | 0 | 0 | 0 | | 3 | 1 | 1 | 0 | | 1 | 0 | 0 | 1 | |
| Non-smoker | 8 | 5 | 3 | 0 | | 2 | 1 | 1 | 0 | | 7 | 2 | 4 | 1 | |
| **Comorbidities** | | | | | | | | | | | | | | | |
| NKMI | 9 | 5 | 4 | 0 | 0.999 | 3 | 1 | 2 | 0 | 0.156 | 3 | 2 | 1 | 0 | 0.614 |
| DM | 0 | 0 | 0 | 0 | | 6 | 4 | 0 | 2 | | 6 | 1 | 3 | 2 | |
| HPT | 0 | 0 | 0 | 0 | | 4 | 3 | 0 | 1 | | 6 | 1 | 3 | 2 | |
| KD | 0 | 0 | 0 | 0 | | 1 | 1 | 0 | 0 | | 2 | 0 | 1 | 1 | |
| **Year of TB infection** | | | | | | | | | | | | | | | |
| 2011–2016 | 0 | 0 | 0 | 0 | – | 2 | 1 | 1 | 0 | 0.477 | 1 | 1 | 0 | 0 | 0.325 |
| 2017–2021 | 0 | 0 | 0 | 0 | | 7 | 4 | 1 | 2 | | 8 | 2 | 4 | 2 | |
| **TB symptoms:** | | | | | | | | | | | | | | | |
| **Shortness of breath** | | | | | | | | | | | | | | | |
| Yes | 0 | 0 | 0 | 0 | – | 3 | 0 | 2 | 1 | 0.034 | 5 | 2 | 3 | 0 | 0.196 |
| No | 0 | 0 | 0 | 0 | | 6 | 5 | 0 | 1 | | 4 | 1 | 1 | 2 | |
| **Fever** | | | | | | | | | | | | | | | |
| Yes | 0 | 0 | 0 | 0 | – | 6 | 3 | 2 | 1 | 0.509 | 9 | 3 | 4 | 2 | 1.000 |
| No | 0 | 0 | 0 | 0 | | 3 | 2 | 0 | 1 | | 0 | 0 | 0 | 0 | |
| **Hemoptysis** | | | | | | | | | | | | | | | |
| Yes | 0 | 0 | 0 | 0 | – | 1 | 1 | 0 | 0 | 0.638 | 2 | 0 | 1 | 1 | 0.413 |
| No | 0 | 0 | 0 | 0 | | 8 | 4 | 2 | 2 | | 7 | 3 | 3 | 1 | |
| **Prolonged cough** | | | | | | | | | | | | | | | |
| Yes | 0 | 0 | 0 | 0 | – | 9 | 5 | 2 | 2 | 1.000 | 9 | 3 | 4 | 2 | 1.000 |
| No | 0 | 0 | 0 | 0 | | 0 | 0 | 0 | 0 | | 0 | 0 | 0 | 0 | |
| **Loss of weight** | | | | | | | | | | | | | | | |
| Yes | 0 | 0 | 0 | 0 | – | 7 | 5 | 2 | 0 | 0.011 | 5 | 2 | 2 | 1 | 0.894 |
| No | 0 | 0 | 0 | 0 | | 2 | 0 | 0 | 2 | | 4 | 1 | 2 | 1 | |
| **Night sweats** | | | | | | | | | | | | | | | |
| Yes | 0 | 0 | 0 | 0 | – | 7 | 3 | 2 | 2 | 0.358 | 0 | 0 | 0 | 0 | 1.000 |
| No | 0 | 0 | 0 | 0 | | 2 | 2 | 0 | 0 | | 9 | 3 | 4 | 2 | |
| **Sputum AFB smear index** | | | | | | | | | | | | | | | |
| +++ | 0 | 0 | 0 | 0 | – | 3 | 1 | 1 | 1 | 0.218 | 3 | 2 | 0 | 1 | 0.367 |

| Characteristics | Healthy control | | | | | TB-SIT745/EAI1-MYS | | | | | TB-Non-SIT745/EAI1-MYS | | | | |
|---|---|---|---|---|---|---|---|---|---|---|---|---|---|---|---|
| | n = 9 | AA | AT | TT | p value | n = 9 | AA | AT | TT | p value | n = 9 | AA | AT | TT | p value |
| ++ | 0 | 0 | 0 | 0 | | 1 | 0 | 1 | 0 | | 1 | 0 | 1 | 0 | |
| + | 0 | 0 | 0 | 0 | | 5 | 4 | 0 | 1 | | 5 | 1 | 3 | 1 | |
| **CXR findings Laterality:** | | | | | | | | | | | | | | | |
| Unilateral | 0 | 0 | 0 | 0 | – | 7 | 4 | 1 | 2 | 0.477 | 9 | 3 | 4 | 2 | 1.000 |
| Bilateral | 0 | 0 | 0 | 0 | | 2 | 1 | 1 | 0 | | 0 | 0 | 0 | 0 | |
| **TB severity** | | | | | | | | | | | | | | | |
| Mild | 0 | 0 | 0 | 0 | – | 3 | 2 | 0 | 1 | 0.663 | 3 | 1 | 1 | 1 | 0.579 |
| Moderate | 0 | 0 | 0 | 0 | | 3 | 1 | 1 | 1 | | 4 | 2 | 2 | 0 | |
| Severe | 0 | 0 | 0 | 0 | | 3 | 2 | 1 | 0 | | 2 | 0 | 1 | 1 | |

Note:
NKMI, No known medical illness; DM, diabetes mellitus; HPT, hypertension; KD, kidney diseases; $p < 0.05$ indicates statistical significance.

**Table 7 Association of IL-10 (-1,082 A/G) SNPs with demographic and clinical parameters in healthy controls and tuberculosis patients infected with SIT745/EAI1-MYS and non-SIT745/EAI1-MYS *M. tuberculosis* strains.**

| Characteristics | Healthy control | | | | | TB-SIT745/EAI1-MYS | | | | | TB-Non-SIT745/EAI1-MYS | | | | |
|---|---|---|---|---|---|---|---|---|---|---|---|---|---|---|---|
| | n = 9 | AA | AG | GG | p value | n = 9 | AA | AG | GG | p value | n = 9 | AA | AG | GG | p value |
| **Gender** | | | | | | | | | | | | | | | |
| Male | 4 | 4 | 0 | 0 | 1 | 7 | 6 | 1 | 0 | 0.076 | 2 | 1 | 1 | 0 | 0.563 |
| Female | 5 | 5 | 0 | 0 | | 2 | 0 | 2 | 0 | | 7 | 6 | 1 | 0 | |
| **Age groups** | | | | | | | | | | | | | | | |
| <50 | 6 | 6 | 0 | 0 | 1 | 3 | 1 | 2 | 0 | 0.32 | 4 | 3 | 1 | 0 | 0.984 |
| >50 | 3 | 3 | 0 | 0 | | 6 | 5 | 1 | 0 | | 5 | 4 | 1 | 0 | |
| **Smoking history** | | | | | | | | | | | | | | | |
| Active smoker | 1 | 1 | 0 | 0 | 1 | 4 | 3 | 1 | 0 | 0.228 | 1 | 1 | 0 | 0 | 0.400 |
| Ex-smoker | 0 | 0 | 0 | 0 | | 3 | 3 | 0 | 0 | | 1 | 0 | 1 | 0 | |
| Non-smoker | 8 | 8 | 0 | 0 | | 2 | 0 | 2 | 0 | | 7 | 6 | 1 | 0 | |
| **Comorbidities** | | | | | | | | | | | | | | | |
| NKMI | 9 | 9 | 0 | 0 | 1 | 3 | 1 | 2 | 0 | 0.829 | 3 | 2 | 1 | 0 | 0.987 |
| DM | 0 | 0 | 0 | 0 | | 6 | 5 | 1 | 0 | | 6 | 5 | 1 | 0 | |
| HPT | 0 | 0 | 0 | 0 | | 4 | 2 | 1 | 0 | | 6 | 5 | 1 | 0 | |
| KD | 0 | 0 | 0 | 0 | | 1 | 1 | 0 | 0 | | 2 | 2 | 0 | 0 | |
| **Year of TB infection** | | | | | | | | | | | | | | | |
| 2011–2016 | 0 | 0 | 0 | 0 | 1 | 2 | 1 | 1 | 0 | 0.851 | 1 | 1 | 0 | 0 | 0.851 |
| 2017–2021 | 0 | 0 | 0 | 0 | | 7 | 5 | 2 | 0 | | 8 | 6 | 2 | 0 | |
| **Shortness of breath** | | | | | | | | | | | | | | | |
| Yes | 0 | 0 | 0 | 0 | 1 | 3 | 2 | 1 | 0 | 1 | 5 | 4 | 1 | 0 | 0.984 |
| No | 0 | 0 | 0 | 0 | | 6 | 4 | 2 | 0 | | 4 | 3 | 1 | 0 | |
| **Fever** | | | | | | | | | | | | | | | |
| Yes | 0 | 0 | 0 | 0 | 1 | 6 | 3 | 3 | 0 | 0.324 | 9 | 7 | 2 | 0 | 1 |
| No | 0 | 0 | 0 | 0 | | 3 | 3 | 0 | 0 | | 0 | 0 | 0 | 0 | |

(Continued)

| Table 7 (continued) | | | | | | | | | | | | | | | |
|---|---|---|---|---|---|---|---|---|---|---|---|---|---|---|---|
| Characteristics | Healthy control | | | | | TB-SIT745/EAI1-MYS | | | | | TB-Non-SIT745/EAI1-MYS | | | | |
| | $n = 9$ | AA | AG | GG | p value | $n = 9$ | AA | AG | GG | p value | $n = 9$ | AA | AG | GG | p value |
| **Hemoptysis** | | | | | | | | | | | | | | | |
| Yes | 0 | 0 | 0 | 0 | 1 | 1 | 1 | 0 | 0 | 0.754 | 2 | 1 | 1 | 0 | 0.563 |
| No | 0 | 0 | 0 | 0 | | 8 | 5 | 3 | 0 | | 7 | 6 | 1 | 0 | |
| **Prolonged cough** | | | | | | | | | | | | | | | |
| Yes | 0 | 0 | 0 | 0 | 1 | 9 | 6 | 3 | 0 | 1 | 9 | 7 | 2 | 0 | 1 |
| No | 0 | 0 | 0 | 0 | | 0 | 0 | 0 | 0 | | 0 | 0 | 0 | 0 | |
| **Loss of weight** | | | | | | | | | | | | | | | |
| Yes | 0 | 0 | 0 | 0 | 1 | 7 | 4 | 3 | 0 | 0.525 | 5 | 3 | 2 | 0 | 0.357 |
| No | 0 | 0 | 0 | 0 | | 2 | 2 | 0 | 0 | | 4 | 4 | 0 | 0 | |
| **Night sweats** | | | | | | | | | | | | | | | |
| Yes | 0 | 0 | 0 | 0 | 1 | 7 | 5 | 2 | 0 | 0.851 | 0 | 0 | 0 | 0 | 1 |
| No | 0 | 0 | 0 | 0 | | 2 | 1 | 1 | 0 | | 9 | 7 | 2 | 0 | |
| **Sputum AFB smear index** | | | | | | | | | | | | | | | |
| +++ | 0 | 0 | 0 | 0 | 1 | 3 | 3 | 0 | 0 | 0.462 | 3 | 1 | 2 | 0 | 0.272 |
| ++ | 0 | 0 | 0 | 0 | | 1 | 0 | 1 | 0 | | 1 | 1 | 0 | 0 | |
| + | 0 | 0 | 0 | 0 | | 5 | 3 | 2 | 0 | | 5 | 5 | 0 | 0 | |
| **CXR findings** | | | | | | | | | | | | | | | |
| Laterality: | | | | | 1 | | | | | 0.076 | | | | | |
| Unilateral | 0 | 0 | 0 | 0 | | 7 | 6 | 1 | 0 | | 9 | 7 | 2 | 0 | 1 |
| Bilateral | 0 | 0 | 0 | 0 | | 2 | 0 | 2 | 0 | | 0 | 0 | 0 | 0 | |
| **TB severity** | | | | | | | | | | | | | | | |
| Mild | 0 | 0 | 0 | 0 | 1 | 3 | 3 | 0 | 0 | 0.557 | 3 | 3 | 0 | 0 | 0.773 |
| Moderate | 0 | 0 | 0 | 0 | | 3 | 2 | 1 | 0 | | 4 | 3 | 1 | 0 | |
| Severe | 0 | 0 | 0 | 0 | | 3 | 1 | 2 | 0 | | 2 | 1 | 1 | 0 | |

Note:
NKMI, no known medical illness; DM, diabetes mellitus; HPT, hypertension; KD, kidney diseases; $p < 0.05$ indicates statistical significance.

## Association of IL-10 (−1,082 A/G) SNPs with demographic and clinical parameters in healthy controls and tuberculosis patients infected with SIT745/EAI1-MYS and non-SIT745/EAI1-MYS *M. tuberculosis* strains

For the analyses of the association between IL-10 (−1,082 A/G) SNPs and the characteristics of healthy controls and TB patients, the findings in Table 7 showed that a majority of the participants possessed the IL-10 (−1,082) AA genotype ($n = 22/27$, 82%), which included all the healthy controls ($n = 9/9$, 100%) and 13 out of 18 TB patients. Whereas, none of the participants carried IL-10 (−1,082) GG genotypes.

Among the six patients with the IL-10 (−1,082) AA genotype in the SIT745/EAI1-MYS group, all were male with a history of smoking. A majority of them ($n = 5/6$, 83%) were over 50 years old and had underlying DM. None of the AA genotype carriers exhibit bilateral consolidation, but one of them experienced hemoptysis and was classified as having severe TB ($n = 1/6$, 17%). Meanwhile, the three IL-10 (−1,082) AG genotype

carriers in this group comprised both females and one male patient. Two of them had severe TB ($n$ = 2/3, 67%) due to bilateral consolidation chest findings.

In the non-SIT745/EAI1-MYS group, the seven IL-10 (−1,082) AA genotype carriers were predominantly females ($n$ = 6/7, 86%), non-smokers ($n$ = 6/7, 86%), and had underlying DM and HPT ($n$ = 5/7, 71%). In terms of signs and symptoms, 57% ($n$ = 4/7) developed shortness of breath, but none of them presented with bilateral chest radiographs. One patient with the IL-10 (−1,082) AA had severe TB with hemoptysis ($n$ = 1/7, 14%). The IL-10 (−1,082) AG carriers in this group comprised one male ($n$ = 1/2, 50%) and one female ($n$ = 1/2, 50%) patient. Besides gender, this group had a balanced distribution of age group, comorbidities, and smoking history. Both patients experienced weight loss and a third-degree AFB smear. One of them was recorded as having severe TB due to hemoptysis ($n$ = 1/2, 50%).

## Association of IFN-γ and IL-10 genetic polymorphisms with demographic and clinical parameters among TB patients

For this analysis, this study aimed to determine the prevalence of dominant AA alleles and the recessive AT/TT or AG alleles for the respective IFN-γ and IL-10 genetic polymorphisms and their association with various demographic and clinical parameters among the TB patients (Table 8).

For IFN-γ +874 SNPs, the dominant AA alleles were prevalent among the males ($n$ = 5/8, 63%), young age group ($n$ = 5/8, 63%), those with smoking history ($n$ = 5/8, 63%) and underlying DM ($n$ = 5/8, 63%). More than half of these IFN-γ (+874) homozygous A carriers ($n$ = 5/8, 63%) were infected by the SIT745/EAI1-MYS *M. tuberculosis* strain. Regarding the TB symptoms, patients with this IFN-γ (+874) homozygous A alleles were less likely to develop shortness of breath ($n$ = 7/8, 88%) and night sweats ($n$ = 5/8, 63%). Most of them experienced weight lost ($n$ = 7/8, 88%) and had unilateral chest findings ($n$ = 7/8, 88%), while 25% ($n$ = 2/8) of these AA allele carriers developed severe TB symptoms.

In contrast, 56% of the patients ($n$ = 10/18), especially females, carried either homozygous or heterozygous T allele at the IFN-γ (+874) SNPs. The majority of these T allele carriers belonged to the elderly group ($n$ = 8/10, 80%) and had comorbidities with DM and HPT. It was also found that 60% of the patients with IFN-γ (+874) T allele had acquired TB due to a non-SIT745/EAI1-MYS *M. tuberculosis* infection. The T allele carriers had also been the most prevalent in developing shortness of breath ($n$ = 6/10, 60%), and 30% ($n$ = 3/10) of them had severe TB infections characterized by hemoptysis and bilateral pulmonary infections.

As for the IL-10 (−1,082) A/G SNPs, 72% of the patients ($n$ = 13/18) carried the homozygous AA allele. This majority comprised predominantly elderly individuals suffering from both DM and hypertension (HPT). There is no significant association between the IL-10 (−1,082) AA carriers and the acquisition of specific *M. tuberculosis* strains. In terms of symptoms, all the IL-10 (−1,082) AA carriers had unilateral pulmonary infections, and a majority of them developed mild and moderate TB symptoms. Only 15% of the IL-10 (−1,082) AA carriers had severe TB with hemoptysis.

**Table 8 Association of IFN-γ and IL-10 genetic polymorphisms with various demographic and clinical parameters among TB patients.**

| Parameters | n = 18 (%) | IFN-γ +874 | | | IL-10 -1,082 | | |
|---|---|---|---|---|---|---|---|
| | | AA n = 8 (%) | AT + TT n = 10 (%) | p value | AA n = 13 (%) | AG n = 5 (%) | p value |
| **Gender** | | | | | | | |
| Male | 9 (50.0) | 5 (62.5) | 4 (40.0) | 0.637 | 7 (53.8) | 2 (40.0) | 0.599 |
| Female | 9 (50.0) | 3 (37.5) | 6 (60.0) | | 6 (46.2) | 3 (60.0) | |
| **Age groups** | | | | | | | |
| <50 | 7 | 5 (62.5) | 2 (20.0) | 0.145 | 4 (30.8) | 3 (60.0) | 0.326 |
| >50 | 11 | 3 (37.5) | 8 (80.0) | | 9 (69.2) | 2 (40.0) | |
| **Smoking history** | | | | | | | |
| Active smoker | 5 | 3 (37.5) | 2 (20.0) | 0.827 | 4 (30.8) | 1 (20.0) | 0.859 |
| Ex-smoker | 4 | 2 (25.0) | 2 (20.0) | | 3 (23.0) | 1 (20.0) | |
| Non-smoker | 9 | 3 (37.5) | 6 (60.0) | | 6 (46.2) | 3 (60.0) | |
| **Comorbidities** | | | | | | | |
| NKMI | 6 | 3 (37.5) | 3 (30.0) | 0.945 | 3 (23.0) | 3 (60.0) | 0.659 |
| DM | 12 | 5 (62.5) | 7 (70.0) | | 10 (76.9) | 2 (40.0) | |
| HPT | 10 | 4 (40.0) | 6 (60.0) | | 8 (61.5) | 2 (40.0) | |
| KD | 3 | 1 (12.5) | 2 (20.0) | | 3 (23.0) | 0 (0.0) | |
| **TB strains** | | | | 0.637 | | | 0.599 |
| SIT745/EAI1-MYS | 9 | 5 (62.5) | 4 (40.0) | | 6 (46.2) | 3 (60.0) | |
| Non-SIT745/EAI1-MYS | 9 | 3 (37.5) | 6 (60.0) | | 7 (53.8) | 2 (40.0) | |
| **Year of TB infection** | | | | | | | |
| 2011–2016 | 3 | 2 (25.0) | 1 (10.0) | 0.559 | 2 (15.4) | 1 (20.0) | 0.814 |
| 2017–2021 | 15 | 6 (75.0) | 9 (90.0) | | 11 (84.6) | 4 (80.0) | |
| **Shortness of breath** | | | | | | | |
| Yes | 8 | 2 (25.0) | 6 (60.0) | 0.188 | 6 (46.2) | 2 (40.0) | 0.814 |
| No | 10 | 6 (75.0) | 4 (40.0) | | 7 (53.8) | 3 (60.0) | |
| **Fever** | | | | | | | |
| Yes | 15 | 6 (75.0) | 9 (90.0) | 0.559 | 10 (76.9) | 5 (100.0) | 0.522 |
| No | 3 | 2 (25.0) | 1 (10.0) | | 3 (23.1) | 0 (0.0) | |
| **Hemoptysis** | | | | | | | |
| Yes | 3 | 1 (12.5) | 2 (20.0) | 0.671 | 2 (15.4) | 1 (20.0) | 0.814 |
| No | 15 | 7 (87.5) | 8 (80.0) | | 11 (84.6) | 4 (80.0) | |
| **Prolonged cough** | | | | | | | |
| Yes | 18 | 8 (100.0) | 10 (100.0) | – | 13 (100.0) | 5 (100.0) | – |
| No | 0 | 0 (0.0) | 0 (0.0) | | 0 (0.0) | 0 (0.0) | |
| **Loss of weight** | | | | | | | |
| Yes | 12 | 7 (87.5) | 5 (50.0) | 0.152 | 7 (53.8) | 5 (100.0) | 0.114 |
| No | 6 | 1 (12.5) | 5 (50.0) | | 6 (46.2) | 0 (0) | |
| **Night sweats** | | | | | | | |
| Yes | 7 | 3 (37.5) | 4 (40.0) | 0.914 | 5 (38.5) | 2 (40.0) | 0.952 |
| No | 11 | 5 (62.5) | 6 (60.0) | | 8 (61.5) | 3 (60.0) | |

| Table 8 (continued) | | | | | | | |
| --- | --- | --- | --- | --- | --- | --- | --- |
| Parameters | n = 18 (%) | IFN-γ +874 | | | IL-10 -1,082 | | |
| | | AA n = 8 (%) | AT + TT n = 10 (%) | p value | AA n = 13 (%) | AG n = 5 (%) | p value |
| Sputum AFB smear index | | | | | | | |
| +++ | 6 | 3 (37.5) | 3 (30.0) | 0.635 | 4 (30.8) | 2 (40.0) | 0.622 |
| ++ | 2 | 0 (0.0) | 2 (20.0) | | 1 (7.7) | 1 (20.0) | |
| + | 10 | 5 (62.5) | 5 (50.0) | | 8 (61.5) | 2 (40.0) | |
| CXR findings | | | | | | | |
| Laterality | | | | | | | |
| Unilateral | 16 | 7 (87.5) | 9 (90.0) | 0.968 | 13 (100.0) | 3 (60.0) | 0.129 |
| Bilateral | 2 | 1 (12.5) | 1 (10.0) | | 0 (0.0) | 2 (40.0) | |
| TB severity | | | | | | | |
| Mild | 6 | 3 (37.5) | 3 (30.0) | | 6 (46.2) | 0 (0.0) | |
| Moderate | 7 | 3 (37.5) | 4 (40.0) | 0.941 | 5 (38.5) | 2 (30.0) | 0.142 |
| Severe | 5 | 2 (25.0) | 3 (30.0) | | 2 (15.3) | 3 (60.0) | |

Note:
NKMI, No known medical illness; DM, diabetes mellitus; HPT, hypertension; KD, kidney diseases; $p < 0.05$ indicates statistical significance.

Of the total 18 patients, only five IL-10 (−1,082) heterozygous G carriers were found, and the majority were young females with no comorbidities ($n = 3/5$, 60%). Additionally, 60% of these carriers had acquired infections due to the SIT745/EAI1-MYS *M. tuberculosis* strain. Fever and weight loss were common among all the G allele carriers, with 60% ($n = 3/5$) of them developed severe TB with bilateral parenchymal consolidation and hemoptysis.

# DISCUSSION

The ongoing evolution of *M. tuberculosis* sub-lineages has emerged as a significant factor influencing the pathogenesis of the disease. Various sub-lineages may exhibit distinct characteristics in terms of virulence, drug resistance, and adaptability to diverse host environments. These traits can affect the bacterium's capacity to cause disease, evade the host immune system, and respond to therapeutic interventions (*He et al., 2022*). Therefore, this study aimed to explore genetic polymorphisms in inflammatory cytokines and their associations with TB infections caused by the Malaysian-specific *M. tuberculosis* strain, SIT745/EAI1-MYS. To our knowledge, this study represents the first investigation into allele variations in host cytokines and their implications for TB risks and severity associated with a specific strain. Moreover, to mitigate the potential for ethnic mismatching, the study focused exclusively on individuals belonging to the Kelantanese Malay ethnic group in Malaysia. This approach would minimize potential confounding effects related to ethnic variation in genetic background and to ensure that the observed associations were more likely attributable to the target SNPs.

A recent meta-analysis study has suggested a potential association between the IFN-γ (+874 A/T) and IL-10 (−1,082 A/G) polymorphisms with TB risk, proposing its utility as a predictive biomarker for susceptibility to TB (*Areeshi et al., 2021*). However, our present

study did not find any significant association between the IFN-γ (+874 A/T) and IL-10 (−1,082 A/G) SNPs and susceptibility to TB or the acquisition of particular *M. tuberculosis* strains. Nevertheless, our descriptive analysis revealed that the IFN-γ (+874) T allele were more prevalent among the TB patients. Regarding *M. tuberculosis* strain distribution, the prevalence of IFN-γ (+874) AA genotypes was higher in TB patients infected with SIT745/EAI1-MYS strains compared to those with non-SIT745/EAI1-MYS strains. Additionally, TT genotypes were exclusively observed among patients in both TB groups but were absent among healthy controls. The higher frequency of the +874 T allele in TB patients was consistent with previous reports from Colombia and Pakistan (*Ansari et al., 2009*; *Henao et al., 2006*). However, contradicting findings have been reported in other studies, where the A allele was found to be more prevalent in patients with active disease, and individuals who were IFN-γ (+874) AA homozygous had an increased risk of developing tuberculosis (*Álvarez et al., 2023*; *Adane et al., 2021*; *Araujo et al., 2017*).

In the context of the IL-10 (−1,082) A/G SNPs, the AA genotypes were predominantly observed in all groups, especially among the healthy control subjects. Notably, the presence of the G allele with AG genotype was exclusively detected in TB patients. Available reports suggest that the AG and GG genotypes of the IL-10 (−1,082) SNP are more commonly found among TB patients across diverse populations (*Meenakshi et al., 2013*; *Wani et al., 2021*). Several studies conducted in various regions, including South Africa, China, and India, have documented a higher prevalence of the GG genotype in TB patients compared to healthy controls (*Ates et al., 2008*). This suggests that the G allele might be associated with an increased risk of developing TB. Nonetheless, a meta-analysis data revealed that there was no association between the IL-10 (−1,082) G/A with TB susceptibility (*Gao et al., 2015*). There has been inconsistency in reports regarding the association between cytokine gene polymorphisms and TB risk, resulting in inconclusive evidence and no definitive indicator for the genetic predisposition to this disease. Furthermore, the frequency of these cytokine alleles and their susceptibility to TB varies significantly among different ethnicities. According to meta-analysis studies, although the IFN-γ (+874) T allele was found to correlate with a decreased risk of tuberculosis in the Caucasian population, the polymorphisms in the IFN-γ (+874) gene do not suggest significant associations with TB risks in the Asian population (*Priyanka, Sharma & Sharma, 2021*; *Areeshi et al., 2021*). The interaction between IFN-γ and IL-10 SNPs may impact the comprehensive immune response in individuals with TB. Maintaining a delicate equilibrium between pro-inflammatory and anti-inflammatory signals is paramount for an efficient immune reaction, and genetic variations in both cytokines have the potential to disturb this equilibrium.

In term of disease severity, our findings indicated that TB patients with the IFN-γ (+874) AA and AT genotypes exhibited more severe TB symptoms. Similar finding was recorded for TB patients with IL-10 (−1,082) AG genotype. The higher prevalence of the IFN-γ (+874) AA and IL-10 (−1,082) AG genotypes among SIT745/EAI1-MYS TB patients corresponded with the number of patients who experienced severe TB observed in this group. While not statistically significant, this trend suggests a potential association worthy of further investigation. The lack of statistical significance may be attributed to the

limited sample size of our study. Biologically, the IFN-γ +874 A allele has been linked to lower IFN-γ production compared to individuals with two copies of the T allele (*Araujo et al., 2017*). This may explain why patients with this SNP tend to experience more severe disease manifestations. On the other hand, the relatively smaller number of patients exposed to severe TB infections in the non-SIT745/EAI1-MYS group could be attributed to the higher frequency of the IFN-γ +874 T allele, which leads to elevated production of IFN-γ and has a protective effect during TB infection. Our results align with studies from Colombia and Pakistan, where TB patients with the IFN-γ TT genotype exhibited minimal to moderate symptoms, indicating a stronger association with the more protective form of TB (*Ansari et al., 2009*; *Henao et al., 2006*). While previous studies have primarily focused on the association of the A allele with TB susceptibility, our results suggest that it may also influence disease severity. Clinically, understanding this relationship could aid in identifying patients who may require more intensive treatment. Future studies with larger cohorts are crucial to confirming these findings and elucidating the underlying biological mechanisms.

The relatively small sample size is acknowledged as a limitation of this study, as it could potentially affect the precision of the data analysis. This limitation was primarily due to several unavoidable factors, including the unique nature of *M. tuberculosis* SIT745/EAI1-MYS strain itself. Although this strain is geographically specific to our region, it constitutes a relatively small proportion (15%) of the overall *M. tuberculosis* strains in Malaysia, compared to the more predominant Beijing lineage (25.5%) (*Ismail et al., 2014*). Additionally, patient recruitment was constrained by factors such as refusal to participate, patient death, loss to follow-up, and exclusion based on study criteria. While the ORs indicated a possible increased risk associated with these SNPs in TB susceptibility, the lack of statistical significance, as reflected by wide confidence intervals, prevent definitive conclusions. However, based on the Cohen's *w* and *post hoc* power analysis $(1 - \beta)$ values for IFN-γ (+874 A/T) and IL-10 (−1,082 A/G), this study demonstrates meaningful large effect sizes and moderate to high statistical power that support the relevance of the findings despite the limited sample size. These results provide preliminary evidence for the potential genetic associations with TB caused by the SIT745/EAI1-MYS strains, in addition to several interesting findings, for instance the distribution of individuals with IFN-γ (+874) TT and IL-10 (−1,082) AG genotypes among TB patients. Further studies involving multi-centers and larger cohorts would be valuable to validate the identified associations and strengthen the contribution of these findings to the broader literature.

Another limitation of this study is the inability to investigate the functional significance of the relationship between those two single SNPs located in the IFN-γ and IL-10 genes and susceptibility to TB. The functional analysis, which would typically involve assessing gene expression levels and cytokine production from peripheral blood, could not be performed in the current study. While useful for genotyping, buccal swab samples used in this study did not provide sufficient material to evaluate cytokine production or gene expression directly. Hence, whether or not the production level of IFN-γ and IL-10 might influence susceptibility or severity in pulmonary TB among Kelantanese Malay ethnic is unknown. Incorporation of blood samples for future studies is essential to allow for

functional analysis of gene expression and cytokine production. This would enable a more comprehensive understanding of the role of these genetic variants in TB pathogenesis.

Beyond that, this study was limited to the analysis of two SNPs, which may overlook other important genetic variations associated with TB risk. This limitation highlights the need for continued investigations on the inclusion of additional cytokine markers beyond IFN-γ and IL-10, such as TNF-α, IL-12, IL-23, IL-27, IL-7 and IL-6, which are also known to play critical roles in the host immune response to *M. tuberculosis* infection (*Cavalcanti et al., 2012*). Expanding the genetic panel in future research may help uncover broader immunogenetic patterns contributing to TB susceptibility and disease progression. Furthermore, future studies with a longitudinal design and integration of clinical treatment outcomes are necessary to assess the potential impact of these SNPs on drug response or treatment success. Due to the important roles of IFN-γ and IL-10 in immune modulation during TB treatment, genetic variations in these cytokines may influence therapeutic outcomes. Such investigations would be crucial in supporting the development of personalized TB therapies based on host genetic profiles. Previous studies reported that the different rates of IL-10 production associated with the IL-10 polymorphism did not affect susceptibility to TB (*Gao et al., 2015*), while IFN-γ concentrations were found lower in the IFN-γ (+874) AA homozygotes than in patients carrying the T allele.

## CONCLUSIONS

Our study reported a higher frequency of the IFN-γ (+874) TT and IL-10 (−1,082) AG genotypes among TB patients compared to healthy controls. This suggests a potential genetic predisposition influencing susceptibility to TB. The IFN-γ (+874) polymorphism is associated with cytokine production levels, which are crucial for an effective immune response against *M. tuberculosis*. The IL-10 (−1,082) AG genotype, known to affect anti-inflammatory cytokine levels, might modulate the immune response, potentially favoring the pathogen's persistence. Additionally, we found that the IFN-γ (+874) AA and IL-10 (−1,082) AG genotypes were more prevalent among TB patients infected with the SIT745/EAI1-MYS *M. tuberculosis* strains, indicating a possible interaction between host genetic variants and strain-specific pathogen characteristics that could influence TB severity.

However, based on the findings, no statistically significant associations were observed between the IFN-γ (+874) A/T (rs2430561) and IL-10 (−1,082) A/G (rs1800896) polymorphisms and the susceptibility or severity of TB in our cohort. Additionally, no significant correlation was found with specific *M. tuberculosis* strains. These current findings suggest that the IFN-γ (+874) TT and IL-10 (−1,082) AG genotypes might not be the primary risk factors for TB in our study population. The interaction between genetic factors and TB risk or severity is complex, influenced by multiple genes, environmental exposures (such as socioeconomic status, lifestyle choices and nutritional intake), and host-related factors (including age, gender and comorbidities). In line with previous studies showing mixed results, our findings underscore the possibility that genetic associations with TB might be population-specific.

Despite the limitations, our study provides valuable insights into the potential role of host genetic factors in TB pathogenesis, particularly in the context of the SIT745/EAI1-MYS strain. Identifying polymorphisms such as those in IFN-γ and IL-10 may contribute to the development of personalized treatment strategies in the future, especially when combined with cytokine expression data and clinical phenotypes. This genetic perspective also supports the inclusion of host factors in vaccine and drug development, paving the way for precision medicine approaches in TB diagnosis, treatment, and control. Further multi-center studies involving larger and more diverse cohorts are warranted to validate these findings and move toward in-depth understanding of host-pathogen interactions in TB.

## ACKNOWLEDGEMENTS

The authors wish to thank the Hospital Universiti Sains Malaysia and Department of Medical Microbiology and Parasitology, School of Medical Sciences, Universiti Sains Malaysia, for providing the facilities used in this study.

### Funding

This research was supported by the Research University Grant (1001.PPSP.8012298) from Universiti Sains Malaysia. The funders had no role in study design, data collection and analysis, decision to publish, or preparation of the manuscript.

### Grant Disclosures

The following grant information was disclosed by the authors:
Research University: 1001.PPSP.8012298.
Universiti Sains Malaysia.

### Competing Interests

The authors declare that they have no competing interests.

### Author Contributions

- Nik Mohd Noor Nik Zuraina performed the experiments, analyzed the data, prepared figures and/or tables, authored or reviewed drafts of the article, and approved the final draft.
- Nur Annisa Nasuha Mohd Sedi performed the experiments, prepared figures and/or tables, and approved the final draft.
- Mohammed Dauda Goni analyzed the data, authored or reviewed drafts of the article, and approved the final draft.
- Suharni Mohamad conceived and designed the experiments, authored or reviewed drafts of the article, and approved the final draft.
- Mohd Nor Norazmi conceived and designed the experiments, authored or reviewed drafts of the article, and approved the final draft.

- Siti Suraiya conceived and designed the experiments, authored or reviewed drafts of the article, and approved the final draft.

## Human Ethics

The following information was supplied relating to ethical approvals (*i.e.*, approving body and any reference numbers):

This study was approved by the Human Research Ethics Committee, Universiti Sains Malaysia (Reference number: USM/JEPeM/18080362).

## Data Availability

The raw data are available in the Supplemental File.

## Supplemental Information

Supplemental information for this article can be found online at http://dx.doi.org/10.7717/peerj.19576#supplemental-information.

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
