# Peer review of "Interferon-gamma and interleukin-10 genetic polymorphisms among the pulmonary tuberculosis caused by Malaysia-specific Mycobacterium tuberculosis strain: SIT745/EAI1-MYS"

_PeerJ, doi:10.7717/peerj.19576_

## Round 0.1 · original submission · Major Revisions

After carefully reviewing your manuscript, I would like to inform you that it requires a major revision before being considered for publication. While the findings are compelling, the limited sample size significantly undermines the conclusions' statistical strength and broader applicability. Nonetheless, this study is relevant to the fields of infectious disease genetics and TB epidemiology. However, it deserves strong consideration for publication, provided that the concerns regarding sample size and the need for additional validation are adequately addressed. Please consider the reviewers’ feedback and comments carefully before submitting a revised version of your manuscript.

·

Basic reporting

In the manuscript “Interferon-Gamma and Interleukin-10 Genetic Polymorphisms among Pulmonary Tuberculosis Caused by the Malaysia-Specific Mycobacterium tuberculosis Strain: SIT745/EAI1-MYS” by Nik Mohd Noor Nik Zuraina et al., the authors aim to investigate the role of IFN-γ (+874) A/T and IL-10 (-1082) A/G SNPs in tuberculosis (TB) disease susceptibility among patients infected with the M. tuberculosis SIT745/EAI1-MYS strain. While the study addresses an important question, there are several significant methodological issues that need to be addressed before it can be considered for publication in PeerJ.

Experimental design

Main concerns:
Sample Size: The sample sizes of the groups are insufficient (n = 9) to reliably detect significant differences. Statistical power analysis should have been conducted prior to the study to ensure that the sample size is adequate. Given that the TB-SIT745/EAI1-MYS lineage is common in Malaysia, it would be more appropriate to analyze larger sample groups to provide more meaningful and generalizable results.
Statistical Tests: The statistical tests used in the manuscript do not align with the study’s primary objective. To properly assess the role of genetic variants in TB susceptibility, the authors should have separately compared the genetic variants in the TB-SIT745/EAI1-MYS group to those in healthy controls and patients infected with non-SIT745/EAI1-MYS strains, ideally using much larger groups. This comparison would provide a more robust framework to interpret the genetic associations.
Clinical Group Matching: The authors state that subjects had no history of immune-related disorders. However,most of their patients have diabetes mellitus, which is known to suppress the immune system. This discrepancy raises concerns regarding the comparability of the groups. It is essential to match the groups by clinical parameters, including diabetes status, to minimize confounding effects.

Validity of the findings

Due to the small sample size, the findings cannot be considered statistically valid.

·

Basic reporting

I have reviewed the manuscript and appreciate the novel approach in investigating the association between IFN-γ (+874) A/T and IL-10 (-1082) A/G polymorphisms with TB susceptibility, particularly in the context of the SIT745/EAI1-MYS M. tuberculosis strain. The study provides valuable insights into host-pathogen interactions and highlights potential genetic markers that could influence TB severity. The methodology, especially the use of allele-specific PCR, is well-executed, and the findings contribute to the growing understanding of TB epidemiology in Malaysia. However, the small sample size limits the statistical power and generalizability of the results. Additionally, functional validation of these SNPs would strengthen the biological relevance of the findings. Expanding the study cohort and incorporating cytokine expression analysis could enhance the impact of this research. Overall, this study presents important preliminary data that can pave the way for further investigations into genetic susceptibility in TB.
Positives:
1. The study explores a novel association between host genetic polymorphisms and M. tuberculosis strain-specific susceptibility.
2. It focuses on the SIT745/EAI1-MYS strain, contributing to the understanding of TB epidemiology in Malaysia.
3. The use of allele-specific PCR provides a reliable method for detecting SNP variations in IFN-γ and IL-10 genes.
4. Findings highlight potential genetic markers that could influence TB severity, paving the way for personalized medicine.
5. The research strengthens the understanding of host-pathogen interactions in TB progression.
Negatives:
1. The small sample size (n=9 per group) limits the statistical power of the findings.
2. Lack of functional validation of SNPs makes it difficult to determine their direct impact on TB pathogenesis.
3. The study does not explore environmental or lifestyle factors that may influence TB severity.
4. A more diverse patient cohort, including different ethnic backgrounds, would strengthen generalizability.
5. The lack of in-depth mechanistic studies limits the biological interpretation of the results.
Lacuna (Gaps in Research):
1. The study lacks an analysis of other cytokines and immune pathways influenced by these SNPs.
2. Functional assays to confirm the effect of these polymorphisms on cytokine expression are missing.
3. The potential impact of these SNPs on TB drug response has not been considered.
4. A larger cohort study is needed to confirm strain-specific genetic associations.
5. The absence of longitudinal data prevents understanding how these SNPs affect TB progression over time.
Limitations:
1. The small sample size restricts statistical significance and generalizability.
2. The study focuses on only two SNPs, overlooking other genetic variations that may influence TB susceptibility.
3. The cross-sectional design does not establish a causal relationship between SNPs and disease severity.
4. Lack of functional validation experiments weakens the biological interpretation of the results.
5. The study does not account for host immune status or comorbidities that may impact TB severity.
Improvements:
1. Expanding the sample size to improve statistical robustness and reliability.
2. Including additional SNPs and cytokine markers to gain a more comprehensive genetic profile.
3. Conducting functional studies to validate how these SNPs influence cytokine production.
4. Integrating environmental and lifestyle factors to assess their role in disease susceptibility.
5. Longitudinal studies to track disease progression in relation to these genetic polymorphisms.
Translational Value:
1. Findings may aid in identifying genetic markers for TB susceptibility and severity.
2. Potential application in developing personalized treatment strategies for TB patients.
3. The study highlights the importance of considering host genetics in TB vaccine and drug development.
4. Understanding strain-specific host interactions can help in TB control and prevention strategies.
5. The research may contribute to precision medicine approaches in TB diagnostics and treatment planning.

Experimental design

The manuscript presents original primary research that aligns well with the aims and scope of the journal. It explores the genetic factors influencing tuberculosis susceptibility, particularly in relation to a specific M. tuberculosis strain, contributing to the understanding of host-pathogen interactions. The study’s focus on genetic polymorphisms and disease severity is relevant to infectious disease research, immunogenetics, and TB epidemiology, making it a suitable submission for the journal.

Validity of the findings

The study provides valuable insights into the genetic factors influencing tuberculosis susceptibility, particularly in relation to the SIT745/EAI1-MYS strain. While the impact and novelty have not been explicitly assessed, the findings contribute to the understanding of host-pathogen interactions. Meaningful replication with a larger cohort would be encouraged to validate the associations and strengthen the study’s contribution to the literature. Clearly stating the rationale and potential benefits of replication would further enhance its significance.

Additional comments

Comments to Authors:
I have reviewed your manuscript and commend the effort in exploring the genetic factors influencing tuberculosis susceptibility, particularly in relation to the SIT745/EAI1-MYS M. tuberculosis strain. The study provides valuable insights into the potential role of IFN-γ (+874) A/T and IL-10 (-1082) A/G polymorphisms in TB severity, contributing to a deeper understanding of host-pathogen interactions. The methodology, including allele-specific PCR, is well executed, and the findings highlight an interesting strain-specific association. However, the small sample size limits the statistical power and generalizability of the conclusions. Additionally, further functional validation of these SNPs and their impact on cytokine expression would strengthen the study’s findings. Expanding the cohort and including additional clinical and immunological parameters could enhance the translational relevance of this research. Overall, the study presents important preliminary data that lays the foundation for future investigations into genetic susceptibility in TB.

---

## Round 0.2 · accepted · Accept

The authors have effectively addressed all the concerns raised by the reviewers in the revised manuscript. They acknowledged the limitation posed by the small sample size and strengthened the statistical analysis. Additionally, the discussion section has been improved to include future research directions and an overview of the study's limitations.

·

Basic reporting

The manuscript is clearly written, with well-structured sections and logical flow. The authors have revised the manuscript to address all editorial and reviewer concerns. They clarified the sample limitations, expanded on the methodology, and improved the discussion by including suggestions for future work.

Experimental design

The study presents a focused design investigating specific cytokine gene polymorphisms in a Malaysian TB strain context. Although the sample size is small, the authors have justified this limitation and conducted appropriate post hoc power analyses. Their methodological responses are satisfactory and demonstrate rigor within the constraints.

Validity of the findings

The authors have acknowledged and addressed the statistical limitations and strengthened their argument by incorporating effect size and power analysis. While the conclusions are preliminary, they are appropriately cautious and provide a solid foundation for future studies. I am satisfied with the revised interpretations.

Additional comments

Thank you for your thoughtful responses to the reviewer comments. Your revisions improve the manuscript's clarity, methodological justification, and discussion of translational value. I encourage you to pursue the suggested functional validation and cohort expansion in future studies. The study provides important insights into host-pathogen genetics in TB.